# Phenolic Compounds Recovery from Pomegranate (*Punica granatum* L.) By-Products of Pressurized Liquid Extraction

**DOI:** 10.3390/foods11081070

**Published:** 2022-04-07

**Authors:** Pamela R. Toledo-Merma, Marianné H. Cornejo-Figueroa, Anabel d. R. Crisosto-Fuster, Monique M. Strieder, Larry O. Chañi-Paucar, Grazielle Náthia-Neves, Héctor Rodríguez-Papuico, Mauricio A. Rostagno, Maria Angela A. Meireles, Sylvia C. Alcázar-Alay

**Affiliations:** 1Universidad Nacional Jorge Basadre Grohmann (UNJBG), Tacna 23001, Peru; ptoledom@unjbg.edu.pe (P.R.T.-M.); andcrisosto@upt.pe (A.d.R.C.-F.); hrodriguezp@unjbg.edu.pe (H.R.-P.); 2Universidad Privada de Tacna (UPT), Tacna 23001, Peru; marianne301096@gmail.com; 3LASEFI-Department of Food Engineering, School of Food Engineering (FEA), University of Campinas (UNICAMP), Campinas 13083-862, SP, Brazil; monique_strieder@hotmail.com (M.M.S.); larry.76728@gmail.com (L.O.C.-P.); grazinathia@yahoo.com.br (G.N.-N.); maam1953@unicamp.br (M.A.A.M.); 4Departamento Académico de Ingeniería Agroindustrial, Universidad Nacional Autónoma Altoandina de Tarma (UNAAT), Junín 12650, Peru; 5Multidisciplinary Laboratory of Food and Health (LabMAS), School of Applied Sciences (FCA), University of Campinas (UNICAMP), Limeira 13484-350, SP, Brazil; mauricio.rostagno@fca.unicamp.br

**Keywords:** pomegranate by-products, phenolic compounds, ellagic acid, punicalagin, PLE

## Abstract

This study aimed to valorize pomegranate by-products (peel and carpelar membranes—PPCM) through their high biological potential for phenolic compounds recovery. The influence of lower temperatures (40 and 60 °C) and pressures (20, 40, 60, 80, and 100 bar) than those generally used in pressurized liquid extraction (PLE) was evaluated through global extraction yield (X_0_), and qualitative and quantitative composition of the phenolic compounds. Chromatographic techniques were used to analyze the two treatments with the highest X_0_. Temperature, pressure, and their interaction had a significant influence on X_0_. The best phenolic compounds extraction conditions were using pressurized ethanol at 60 °C and 40 bar (extract 1—E1, 37% on d.b.) and 60 °C and 80 bar (extract 2—E2, 45% on d.b.). Nevertheless, E1 presented a significantly higher content of α, β punicalagin, and ellagic acid (48 ± 2, 146 ± 11, and 25.6 ± 0.3 mg/100 g, respectively) than E2 (40 ± 2, 126 ± 4, and 22.7 ± 0.3 mg/100 g). Therefore, this study could validate the use of low pressures and temperatures in PLE to recover phenolic compounds from pomegranate residues, making this process more competitive and sustainable for the pomegranate industry.

## 1. Introduction

In Latin America and the Caribbean, more than 36 million tons of food loss and waste were recorded at the manufacturing stage in 2019 [1]. Agro-industrial waste generated worldwide represents a considerable fraction, mainly from the fruit and vegetable sector with 45% [2]. These wastes can represent a significant waste of resources such as water, energy, land, labor, and capital. They can also represent a problem of environmental pollution through the emission of greenhouse gases, contributing to global warming and climate change, as well as the appearance of biological hazards (insect pests, bacteria). In the framework of meeting the targets of 12.3 Sustainable Production and Consumption of the Sustainable Development Goals (SDGs) and the circular economy approach, it has been proposed to reintroduce food waste into the production chain by converting it into raw material and adding its value, considering that it is a rich source of bioactive compounds. These compounds are secondary metabolites that are characterized by their multiple beneficial properties for health. Within the fruit sector, pomegranate (*Punica granatum* L.) stands out for having numerous bioactive compounds of high biological value in its different parts: peel, carpelar membranes, arils, husks, and seeds [3,4,5]. The peel and carpelar membranes are agro-industrial by-products obtained from the pomegranate industry and are rich sources of phenolic compounds, which represent about 40–50% of the whole fruit weight [6,7]. Among the main phenolic compounds are phenolic acids (hydroxybenzoic and hydroxycinnamic), flavonoids (anthocyanins, flavonols), tannins (gallotannins and ellagitannins) such as punicalagin isomers, ellagic acid and its derivatives [8,9]. These phenolic compounds have been associated with protection against noncommunicable chronic diseases because of their antioxidant [10,11], anti-cancer [12], antiaging, antimicrobial [13], anti-inflammatory, antitumor [14], antimutagenic, and other properties. Therefore, these compounds can be used in food [15,16,17] and pharmaceutical [18] industries. The recovery of bioactive compounds from pomegranate residues could contribute significatively to improving food security, avoiding food waste, and ensuring economic, environmental, and social benefits [19].

Different extraction technologies have been applied to obtain extracts rich in phenolic compounds. In order to increase the efficiency of the food system and reduce agro-industrial waste, it is necessary to invest in innovation and technologies that can be environmentally friendly. Thus, in recent decades, some clean or green extraction technologies have been developed that do not represent health risks because they use generally recognized as safe (GRAS) solvents such as CO_2_, ethanol, water, glycerol, isopropanol, or their mixtures. Some studies have been carried out for the extraction of bioactive compounds from pomegranate by-products using different technologies such as high-pressure extraction (HPE), ultrasound-assisted extraction (UAE), supercritical fluid extraction (SFE), pressurized liquid extraction (PLE), pulsed electric field (PEF), high-voltage electrical discharges (HVED), high hydrostatic pressure (HHP), and others [20,21,22,23,24,25]. These alternative technologies have significant advantages, such as lower solvent usage (<150 mL), shorter processing time (<75 min), and better extraction yields compared with conventional extraction techniques such as maceration and Soxhlet, which can take up to 24 h using a larger volume of solvent [22]. 

PLE also referred to as accelerated solvent extraction (ASE), subcritical solvent extraction (SSE), and pressurized hot liquid extraction (PHLE), is considered an emerging and sustainable extraction technique. This technique normally employs solvent at high pressures (100–200 bar) and moderate to high temperatures (100–220 °C) [26,27,28]. Pressure and mainly temperature changes have been shown to have an important influence not only on the global extraction yield but also on the selectivity of the phenolic compound extracts. However, the use of high pressures generates a high energy expenditure and thus a high production cost to obtain the extracts [29]. Additionally, the use of high temperatures can cause degradation of thermolabile phenolic compounds, which is a disadvantage to obtaining ingredients or food additives with high antioxidant potential for the food or pharmaceutical industry [24]. PLE has been employed to extract mostly polar but also nonpolar phenolic compounds from agro-industrial wastes such as peels [30], seeds [31], leaves [32], pomace [33], and other food by-products [34]. However, there are few studies on the recovery of phenolic compounds from pomegranate by-products (peels and carpelar membranes as a whole) obtained by the pomegranate juice/arils industry using GRAS solvents at lower pressures and temperatures than those normally applied by PLE, which can allow significant energy savings and preserve better the target phenolic compounds. It could be valuable for pomegranate industrial production and may allow pomegranate processing companies to become more competitive, profitable, and sustainable through time [35,36].

Therefore, this research aimed at the recovery of target phenolic compounds from pomegranate by-products by the application of moderate temperatures and pressures in PLE technology to encourage the valorization of the agro-industrial residues and promote the sustainability of the pomegranate industry processing. The effects of both parameters were evaluated on the global extraction yield, and the chemical composition of pomegranate peel and carpelar membranes (PPCM) extracts was evaluated through chromatographic techniques.

## 2. Materials and Methods

### 2.1. Sample Preparation

The Wonderful pomegranate variety used in this study was sourced from the Ite district (Jorge Basadre province, Tacna, Peru). The PPCM were separated from arils and subjected to a drying process at 40 °C for 72 h in an experimental tray dryer (Tacna, Peru). Once dried, they were grounded in a knife mill (Marconi, model MA340, Piracicaba, Brazil), sieved through a set of sieves (WS Tyler, Wheeling, IL, USA) from 9 (2 mm) to 80 mesh (0.180 mm). The finest particles collected in the bottom pan were separated. The average particle diameter was calculated as 0.50 mm according to the methodology ASAE S424.1 MAR1992. The particles were stored in zip lock high-density polyethylene plastic bags wrapped in aluminum foil and kept in the freezer at −20 °C until later experimental procedures.

### 2.2. Reagents and Standard

All chemicals were of analytical grade. Ethyl acetate (99.8%), glacial acetic acid (99.7%), and methanol (99.8%) were obtained from Dinâmica Co. (São Paulo, Brazil). The formic acid (98%) was purchased from Merck (Darmstadt, Germany). CIAL (São Paulo, Brazil) and Anidrol (São Paulo, Brazil) sourced the HPLC/UV acetonitrile (99.9%, J.T. Baker) and ethanol (99.5%), respectively. MilliQ water was obtained through a purification system (Milli-Q, Millipore, Bedford, MA, USA). The ellagic acid (EA) standard (≥95%, lote # BCBP8742V, Dorset, UK) was purchased from Sigma-Aldrich.

### 2.3. PPCM Proximate Composition

The proximate composition of the PPCM was determined by the moisture, ash, protein, lipid, and total carbohydrate content. Moisture and ash were determined according to the methods 925.09 (AOAC, 1997) and 923.03 (AOAC, 1995), respectively. The proteins were quantified using the method described by Bradford [37] with some modifications [38]. The lipids content was analyzed by the Soxhlet method [39], using petroleum ether as solvent. The remaining substances were assigned to carbohydrates. All analyses were performed in triplicate.

### 2.4. Extraction Procedure by PLE

The extractions were performed in an experimental equipment (Figure 1) following the method described by Santos et al. [40] with some modifications. This equipment consisted of a solvent reservoir, an HPLC pump (Jasco Corporation, model PU-2080, Tokio, Japan) to pump the solvent, a manometer to measure the pressure, and a 6.57 mL extraction vessel, which was connected to a temperature controller. This cell was placed inside an electric heating jacket. In addition, the equipment had two blocking valves and a back pressure valve to maintain constant pressure during extraction. The system connections were made with stainless steel tubing (1/16″ and 1/8″). The effects of PLE temperature (40 and 60 °C) and pressure (20, 40, 60, 80, and 100 bar) on the extraction yield of extracts obtained were studied. For that, a 2 × 5 randomized full factorial design was used, totaling 10 treatments and 20 responses since all treatments were performed in duplicate. The particles of PPCM already prepared (6 g on a d.b.) were placed inside the extraction cell (6.57 mL) of the PLE equipment and heated by a heating jacket until reaching the desired temperature. Then, the system was pressurized with the solvent using an HPLC pump. After reaching the desired pressure by closing the solvent outlet with the blocking valve, the pressure was kept for a static extraction time of 10 min until reaching the system equilibrium. Immediately afterward, the blocking and back-pressure valves were opened and carefully adjusted to maintain the required system pressure. Then, the dynamic extraction started by pumping the solvent, which penetrated throughout the vegetal matrix, extracting solvent-soluble biocompounds at a constant flow rate of 1 mL/min. The dynamic extraction time was around 76 min, and the solvent-to-feed ratio (S/F) was 10. The ethanolic solution containing the PPCM extract was collected in a volumetric glass flask at room temperature. The ethanol was evaporated using a rotary evaporator under a vacuum at 40 °C in the absence of light. The extracts were placed in a desiccator until reaching constant weights. Afterward, the extracts were protected from light to prevent the degradation of the compounds. Thus, they were stored at freezing temperature for further analysis. 

### 2.5. Extract Characterization

#### 2.5.1. Global Extraction Yield

The global extraction yield *X_0, S/F=10_* was calculated according to Equation (1). Where *m_extract_* is the extracted solids mass on a dry basis at the solvent-to-feed (*S/F*) ratio of 10 for a given pressure and temperature and m_PPCM_ is the initial mass of PPCM on a dry basis,
(1)$$X_{0,\frac{S}{F}=10\left(\%\right)}=\frac{m_{extract}}{m_{PPCM}\times 100}$$
the two treatments with the highest global extraction yield were analyzed by chromatography.

#### 2.5.2. Qualitative Analysis of Phenolic Compounds 

A total of 0.3 g of extract was dissolved in 5 mL of methanol. The extract was completely dissolved by a sonication treatment at room temperature for 10 to 15 min in a 135 W ultrasound bath equipment (Ultrasonic clean, Maxiclean 1400, Unique, Indaiatuba, São Paulo, Brazil). Afterward, the solution was filtered through a 0.22-μm nylon syringe filter (Analítica, São Paulo, Brazil) to be analyzed by chromatography.

Previously dissolved extracts were analyzed by thin-layer chromatography (TLC) using the method described by Wagner & Bladt [41] to identify phenolic compounds. Analyses were conducted using 0.20 mm silica gel plates precoated with aluminum 10 × 10 cm (Alugram^®^, Xtra SIL G, Macherey-Nagel, Düren, Germany) with sensitivity UV-266 nm. The EA standard solution was prepared from 3.5 mg of EA in 5 mL of methanol, which was sonicated for 10 min.

Test solutions (10 μL) and EA standard solution (10 μL), previously loaded in a 10 µL glass syringe, were applied on 15 mm points, 15 mm from the lower edge of the plate. The mobile phase was a mixture of ethyl acetate, water, glacial acetic acid, and formic acid (68:18:7:7, *v*/*v*/*v*/*v*). The plate was developed over a distance of 80 mm from the lower edge using a glass chamber saturated for 20 min with the mobile phase. After development, the plate was dried inside a fume extraction hood for 15 min. Then, the plate was analyzed under visible light, UV-light at 254 and 366 nm (Multiband UV-254-366 nm, UVGL-58, Mineralight^®^ Lamp, Upland, CA, EUA). The plate was sprayed with Natural Product (NP) reagent solution (0.5 g of 2-aminoethyl diphenylborinate in 50 mL of methanol) and then dried for 15 min to visualize better the separation of the compounds. Digital images were taken with a 25 Mpx smartphone (Samsung, SM-A505G, Seul, South Korea).

#### 2.5.3. Quantitative Analysis of Punicalagins and Ellagic Acid

The target phenolic compounds were quantified using the method described by Sumere et al. [24] and Qu et al. [42] with some modifications. The extracts were dissolved in water and filtered in a 0.22-μm nylon syringe filter (Analítica, São Paulo, Brazil). Chromatographic separation was performed by HPLC on a Waters Alliance separation module (model 2695D, Milford, CT, USA) equipped with a photodiode array detector (PDA). The individual compounds in the extracts were separated on a Kinetex core-shell technology C18 column (100 × 4.6 mm, 2.6 µm, Phenomenex, Torrance, USA) maintained at 50 °C using a flow rate of 1.6 mL/min. The mobile phase used was type 1 ultrapure acidified water (1% formic acid) (solvent A) and acidified acetonitrile grade HPLC (1% formic acid) (solvent B). The gradient of 0 min: 0% B, 4 min: 4% B, 6 min: 10% B, 10 min: 20% B, 14 min: 50%, 17 min: 90%, 19 min: 90%, and at 20 min: 0% was used. The injection volume was 10 µL. The software used for equipment control and data acquisition was Empower Pro 2 (Waters, Milford, MA, USA). The identification of punicalagin (α and β) and EA was performed by comparing retention times and absorption spectra (UV–Vis) with the standard of EA. The UV absorbance was monitored at 340 nm. The standard calibration curve was obtained by plotting concentrations of 1 to 100 mg/L of EA against the peak area. Regression equations and determination coefficients (R^2^ = 0.9996) were calculated.

### 2.6. Statistical Analysis

Statistical analysis of the experimental results was performed using MINITAB^®^ 18 (Minitab, LLC., PA, USA) software to evaluate the analysis of variance (ANOVA) and apply Tukey’s test to verify the effect of studied variables and significant differences at a confidence level of 95% (*p* < 0.05). Simple effects analysis was carried out using SAS 9.0 System.

## 3. Results

### 3.1. PPCM Proximate Composition

PPCM (pomegranate peel and carpelar membranes) were constituted of 5.7 ± 0.5%, 5.0 ± 0.1%, 4.33 ± 0.02%, and 84.9% on a dry basis of lipids, proteins, ash, and total carbohydrates, respectively. Pomegranate by-products are considered a good source of minerals, in agreement with the results found in PPCM, which were similar to what was reported in a previous study that analyzed pomegranate peel [43]. An important parameter to highlight is the lipid content present; although the pomegranate peel itself is not a representative source, but rather the seeds [44], the content in PPCM was 3 times higher than findings reported in another study [45]. Moreover, PPCM presented high content of total carbohydrates composed of soluble and insoluble fiber and sugars [46]. These composition differences may be due to the inclusion of carpelar membranes in this study, pomegranate variety and origin, climate and soil conditions, the position of the fruit within the tree, harvest time, ripening stage, and other factors [47,48].

### 3.2. PLE Effects on the Global Extraction Yield 

Global extraction yields obtained through PLE are shown in Table 1. The interaction between temperature and pressure presented a significant effect on the global extraction yield (*p* < 0.05, R^2^ = 90.05). The extract obtained by treatment 9 at 60 °C and 80 bar presented the highest extraction yield (X_0_ = 45%). 

Additionally, the global extraction yields acquired (27% to 45%) were higher than those obtained in other studies using green and conventional methods from pomegranate peel [22,49,50,51,52], similar to those reported by Çam & Hişil [35] and lower than other research results [53,54]. These differences could be associated with the inclusion of carpelar membranes in this study, extraction technique, solvent type, pomegranate variety, and origin [3], which are shown in Table 2.

#### 3.2.1. Pressure Effects on the Extraction Yield 

Pressure presented a significant effect on the extraction yield (*p* < 0.05), showing a polynomic tendency. This could be influenced by the extraction solvent (ethanol), which increases its polarity as pressure rises [57]. According to Tukey’s test, the average extraction yields obtained at 40 (34.14%) and 80 bar (38.19%) were significantly different from the average extraction yield at 20 bar (28.71%); however, according to simple effect analysis, 80 bar was the most significant pressure on extraction yield (*p* < 0.05). The drop in extraction yield at 100 bar may be because high pressures could hinder the surface contact between vegetal matrix and solvent. In addition, the path of the solvent can be negatively affected by the compaction of the raw material in the extraction bed promoted by increasing the pressure, leading to a decrease in extraction yield [58,59]. On the other hand, the pressure effect of this study disagrees with the findings of other studies, where the pressure did not have a significant and positive influence on the global extraction yield [29,60]. Alcázar-Alay et al. (2017) evaluated pressures of 20 and 80 bar on the overall extraction yield in order to extract anthocyanins from açai pulp; however, such a range of pressures had neither a positive nor a negative effect on the extraction process.

#### 3.2.2. Temperature Effects on the Extraction Yield

Temperature presented significant effect on the extraction yield (*p* < 0.05). The highest temperature (60 °C) favored the extraction of phenolic compounds. Treatments obtained at this temperature showed a higher extraction yield than those obtained at 40 °C. It agrees with a study that used the same extraction method and showed that temperature influenced significantly the recovery of pomegranate total phenolic compounds [36]. Higher temperatures allow better sample solvation, lower surface tension between the solvent and the plant matrix, and lower viscosity of the solvent. These conditions favor the penetration of the solvent into the matrix and improve the mass transfer of the phenolic compounds from the pomegranate by-products to the solvent [26,27,61]. However, a study demonstrated that temperatures above 60 °C promote the degradation of phenolic compounds such as punicalagin isomers [24].

### 3.3. Phenolic Compounds Identification

Figure 2 presents the results of the thin-layer chromatography in order to compare qualitatively extracts E1 and E2. The presence of ellagic acid was observed clearly in the plates by a light gray band at the same retention factor (Rf = 0.83) as observed for the ellagic acid pattern (Figure 2a,b). Additionally, the ellagic acid polarity may be classified as a medium-low polarity compound because the compound was entrained for a considerable distance by a mobile phase (ethyl acetate: water: glacial acetic acid: formic acid, 68:18:7:7, *v*/*v*/*v*/*v*) whose calculated polarity index was 5.67 [62]. The presence of other phenolic compounds of different degrees of polarity and colors was observed in visible light between the Rf of 0.09 and 0.83 (Figure 2b). When the plate was observed in short wavelength UV light (254 nm) without NP reagent (Figure 2c), the presence of other compounds was better visualized by the separation of bands that were not clearly appreciable in visible light. However, applying NP reagent at 254 nm, the compounds shown became blurred (Figure 2d). With respect to the visualization of the plate in UV light at the long wavelength (366 nm) using NP, the presence of seven bands with different degrees of polarity and coloration was observed in both extracts (Figure 2f). The bands of bright fluorescent coloration may be flavonoids [63]. NP improved the visualization of the phenolic compounds. Thus, pomegranate by-products extracts presented a diversity of phenolic compounds, mostly of polar and medium polar nature.

### 3.4. Punicalagins and Ellagic Acid Content

The extracts E1 and E2 presented a total of 19 phenolic compounds according to HPLC-PDA results. Representative chromatograms of the pomegranate extracts are shown in Figure 3. Six compounds were observed in significant amounts, and three of them were identified and quantified [35,64,65].

The most intense peak was observed for β-punicalagin (6.4 min), followed by two unknown compounds. At 4.8 min, the peak of α-punicalagin was observed, followed by another unknown compound, and then ellagic acid (9.3 min), derived from the hydrolysis of punicalagin [66]. Table 1 shows the target phenolic compound content in E1 and E2. The same phenolic compounds were observed for both extracts; however, E1 presented significantly (*p* < 0.05) higher content of the α, β-punicalagins, and ellagic acid than E2.

Meanwhile, at lower pressure of 40 bar, the punicalagin and ellagic acid were better recovered from the vegetal matrix. This may be because, when the extraction pressure increases, the extraction fluid viscosity and density increase, avoiding the fluid can penetrate better into the plant matrix and interact with the bioactive compounds [67]. In addition, according to Santos et al. [40], who optimized a PLE process where an evaluated range of pressures (50–100 bar) was used in order to recover phenolic compounds from jabuticaba skins, the use of high pressures had a negative effect on the recovery of phenolic compounds, and for that reason, the optimized pressure was set at a range of 48–50 bar. Therefore, it is recommended to work with low pressures depending on the nature of the target compounds to be extracted not only because their recovery may be higher but also because the PLE process becomes more profitable and commercially competitive.

The highest content of α, β—punicalagins, and ellagic acid obtained in the present study was 194.96 mg/100 g and 24.91 mg/100 g, respectively, representing 45% of total phenolic compounds present in PPCM. The tendency for pomegranate residues to have higher punicalagin content compared with ellagic acid was shown in many other studies as well, except for the study where very high pressures of 3820 bar were applied, the ellagic acid content was reported to be considerably higher than punicalagin [49]. Moreover, the amount of ellagic acid reported in the present study was similar to a study where this phenolic compound was recovered (20.7 mg/100 g) from pomegranate peel by ultrasound-assisted extraction [55]. The punicalagin content values were lower than those reported by other authors, as it is shown in Table 2. These differences in the target compound content could depend on the fruit variety [68], preparation technique of the plant matrix, type of extraction solvent, and operation conditions [3,61]. Furthermore, the drying process that was used before the extraction process could influence the quantification of the compounds of interest. For instance, in this study, pomegranate by-products were dried by hot air at 40 °C, while another study using freeze-drying pomegranate peels preserved better the phenolic compounds [69]. Therefore, there are many factors to take into account prior to the recovery of phenolic compounds from food waste.

## 4. Conclusions

In this study, PLE was demonstrated to be a promising and efficient green extraction technology of phenolic compounds from PPCM using ethanol as the extraction solvent. Both evaluated factors, process temperature and pressure and their interaction, significantly influenced the global extraction yield (*p* < 0.05). The best results achieved were obtained using an extraction process at 60 °C and 80 bar. However, using 40 bar at the same temperature, higher α, β—punicalagins, and ellagic acid content were obtained. Therefore, it was possible to enhance this process by validating the low pressures and temperatures with consequent energy savings and the reduction of manufacturing costs. This study is a contribution that suggests the pomegranate processing industry may become a competitive and sustainable industry that could invest in an innovative eco-friendly technology that allows the reintroduction of agro-industrial waste in the production chain, obtaining rich extracts in phenolic compounds such as punicalagin and ellagic acid from pomegranate by-products that can be applied as natural ingredients with antioxidant, preservative, and functional capacity.

## 5. Patents

There is a patent resulting from the work reported in this manuscript that is in process [N° 002066-2020].

## Figures and Tables

**Figure 1 foods-11-01070-f001:**
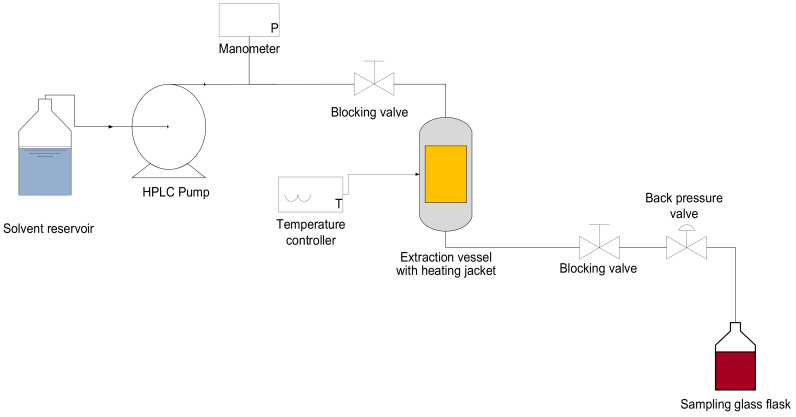
Schematic diagram of the homemade PLE equipment.

**Figure 2 foods-11-01070-f002:**
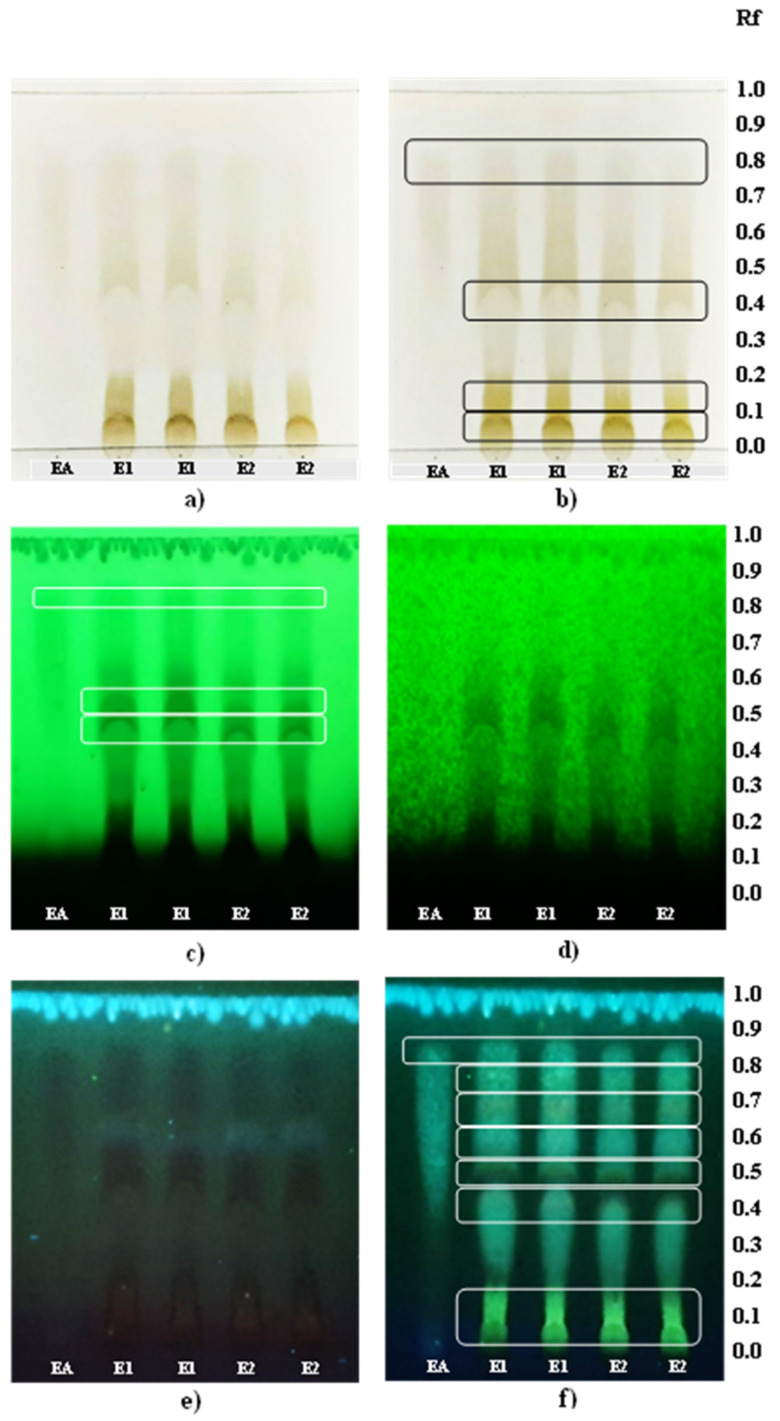
TLC analysis of extracts E1 (40 bar) and E2 (80 bar) under (**a**) visible light without NP, (**b**) visible light with NP, (**c**) UV light at 254 nm without NP, (**d**) UV light at 254 nm with NP, (**e**) UV light at 366 nm without N, and (**f**) UV light at 366 nm with NP.

**Figure 3 foods-11-01070-f003:**
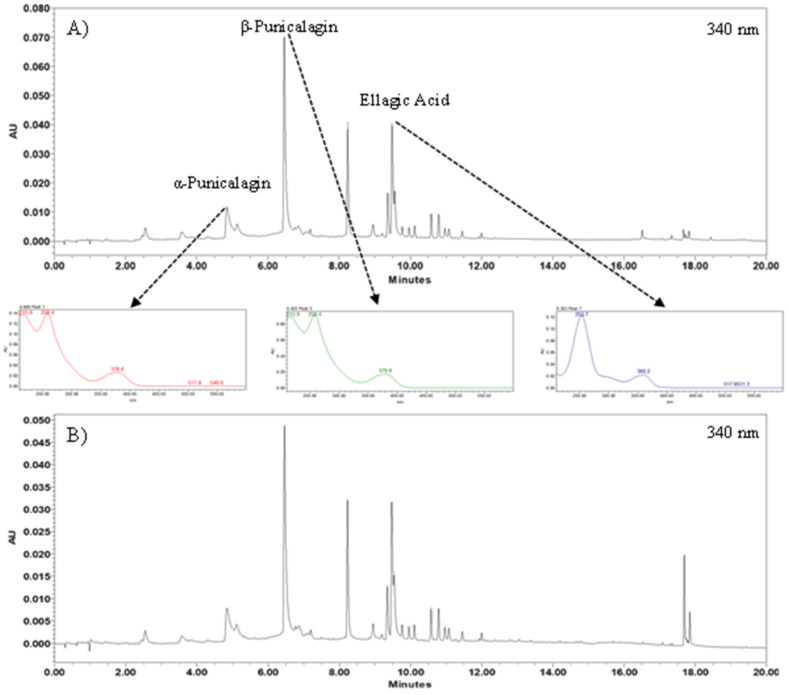
Representative chromatograms at 340 nm of the (**A**) extract 1 and (**B**) extract 2 obtained with the highest extraction yield.

**Table 1 foods-11-01070-t001:** Global extraction yields (X_0_) and PPCM target phenolic compounds content obtained assessing temperature and pressure by PLE.

Treatments	Temperature (°C)	Pressure (bar)	X_0_	Pun α	Pun β	EA
			(%)	(mg/100 g dw)	(mg/100 g dw)	(mg/100 g dw)
1	40	20	27.41 ± 0.77 ^c^			
2	40	40	31.01 ± 1.98 ^b,c^			
3	40	60	30.59 ± 0.38 ^b,c^			
4	40	80	31.39 ± 0.36 ^b,c^			
5	40	100	31.12 ± 3.21 ^b,c^			
6	60	20	30.00 ± 1.96 ^b,c^	48.22 ± 2.05 ^a^	146.58 ± 11.20 ^a^	25.57 ± 0.27 ^a^
7	60	40	37.28 ± 0.41 ^a,b^			
8	60	60	35.36 ± 1.18 ^b,c^	40.41 ± 1.56 ^b^	125.72 ± 3.65 ^b^	22.74 ± 0.30 ^b^
9	60	80	44.99 ± 1.81 ^a^			
10	60	100	36.06 ± 1.31 ^a,b,c^			

Pun α—punicalagin α, Pun β—punicalagin β, EA—ellagic acid; Values of X_0_ are mean ± standard deviation (n = 2); Values of Pun α, Pun β, and EA are mean ± standard deviation (n = 4); Means within a column with different letters (^a–c^) are significantly different (*p* < 0.05).

**Table 2 foods-11-01070-t002:** Extraction of phenolic compounds from pomegranate peel applying green and conventional extraction methods.

Extraction Method	Pomegranate Variety (Origin)	Extraction Solvent	Operation Conditions	Extraction Yield (%)	Target Phenolic Compounds Content (mg/g dw)	Reference
High-Pressure Extraction (HPE)	(Portugal)	36% ethanol	P = 3820 bar Ө = 30 min	24.9–31.3%	3.12 ± 0.4 (α Pun) 3.62 ± 0.4 (β Pun) 691 ± 115 (EA)	[49]
Microwave-Assisted Extraction (MAE)	(Rodi Hellas, Greece)	50% ethanol	S/F = 60/1 MP = 600 W	-	143.64 (α, β Pun)	[20]
Ultrasound-Assisted Extraction (UAE)	Malas (Isfahan, Irán)	70% ethanol	UIL = 105 W/cm^2^ Duty cycle = 50% (10 min) OM = pulse	26.8–41.6%	128.02-146.61 (α, β Pun) 10.12-22.53 (EA)	[51]
	(Beirut, Lebanon)	Water	UP = 400 W Ө = 7 min T < 2 °C	-	0.207 (EA)	[55]
Ultrasound-Assisted Extraction (UAE)	Sishekape-Ferdos	Water	A = 60% Ө = 6.2 min	13.1%	-	[22]
	Wonderful (Apulia, Italy)	70% ethanol	A = 50–80% Ө = 10 min T = 45–70 °C	-	≈40 µg/mL (EA)	[25]
	(Do, Bosnia and Herzegovina, Serbia)	59% ethanol	Ө = 25 min S/F = 44 T = 80 °C	-	11.65 ± 0.42 (EA) 2.87 ± 0.11 (GA) 18.05 ± 0.62 (α, β Pun)	[56]
	Molar (SP, Brazil)	70% ethanol	T = 50–60 °C F = 37 KHz OM = normal and pulse	-	14.8–16.19 (α Pun) 22.45–24.29 (β Pun) 2.13-2.23 (EA)	[53]
Accelerated Solvent Extraction (ASE)	(Turkey)	Water	T = 40 °C P = 103.5 bar	43.3± 2.7%	116.6 ± 12.2 (α, β Pun) 1.25 ± 0.2 (EA)	[35]
	Wonderful (Atacama, Chile)	77% ethanol	T = 200 °C P = 103.4 bar Ө = 20 min	-	17 ± 3.6 (α, β Pun)	[36]
	Wonderful (California, USA)	70% ethanol	T = 60 °C P = 100 bar	-	4.14 ± 0.19 (α Pun) 8.12 ± 0.28 (β Pun) 1.28 ± 0.09 (EA)	[24]
Supercritical Fluid Extraction (SFE)	Wonderful (Elqui valley, Chile)	scCO_2_: ethanol (80:20)	T = 40–50 °C P = 200–300 bar	0.2–8.5%	4–95 (α, β Pun)	[50]
Conventional Extraction	Wonderful (California, USA)	Ethanol	T = 40 °C P = 1.01 bar S/F = 15 Ө = 240 min	17.71%	-	[52]
	(Himachal Pradesh, India)	60% ethanol	T = 50 °C P = 1.01 bar S/F = 30 Ө = 45 min	40–68%	-	[54]

T—extraction temperature, P—extraction pressure, Ө—extraction time, S/F—solvent-to-feed ratio, MP—microwave power, UIL—ultrasonic intensity level, UP—ultrasound power, A—amplitude, F—frequency, OM—operation mode, Pun—punicalagin, EA—ellagic acid, GA—gallic acid.

## Data Availability

Data is contained within the article.

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
