# Peer review of "Phenolic Compounds Recovery from Pomegranate (Punica granatum L.) By-Products of Pressurized Liquid Extraction"

_foods, 2022, doi:10.3390/foods11081070_

Round 1

Reviewer 1 Report

It is opinion of the reviewer that this paper needs several corrections. My individual comments are listed below.

3 – It should be „Liquid”.

7-12 – Authors’ e-mail addresses and their initials must be added.

17 – Full name of “PLE”.

17, 221 and other lines – What does it mean “global extraction”?
19 – Interaction between temperature and time?

38-39 – It must be rephrased.

43 – It must e “Punica granatum …  (with italic).

52 – Difference between “ant-cancer” and “anti-tumor”?

&0 – It must be “Soxhlet”.

164 – Power of this bath?

168 – Silica gel particle size” The layer thickness?

172 – How was loaded?

124, 136 – The duplicated information.

196 – It should be “UV-Vis”.

198 – “R2” is a determination coefficient.

208 – It should be “were constituted”.

Table 1 Why the content of phenolic compounds is reported only for two treatments?

261 – It should be “Tukey’s”.

341 – It should be “…. Chromatograms at 340 nm …”.

398/407 – The initials not full names.

434, 440, 471, 479, 483, 514, 519, 579 – The Latin names must be written with italic.

Author Response

Manuscript ID: foods-1652443

We would like to thank the reviewers of our paper for their helpful feedback to the improvement of this manuscript. We have taken into account all the concerns raised and we have made the suggested modifications.

Reviewer 2 Report

The present study covers the research on the promotion of the no-waste technologies and the extraction of phenolic compounds from pomegranate by-products. According to my opinion, the new obtained knowledge can be useful for the industrial production of added-value phenolic containing ingredients. To improve the quality of the manuscript, I have certain questions as outlined below and suggest a few corrections:

  • In the introduction section please highlight the novelty of the study. There are a lot of studies performed in the area of PLE and Punica granatum. You should clearly indicate the relevancy and novelty.
  • Please explain the abbreviation PPCM in the first mention in the introduction section.
  • Line 97. the peels were separated from arils, not fruits. The whole is fruits that consist of peel (exocarp and mesocarp) and arils (seed with fleshy outer layer).
  • “S/F ratio” Please clarify.
  • Line 198. R2 is the coefficient of determination. Please change.
  • Line 185. The HPLC or UPLC analysis was performed? The columns and conditions seem to be UPLC.
  • Line 200. Please add a description of the regression analysis performed in the statistical section.
  • To what treatments (according to table 1) does the E1 and E2 correspond? Was the phytochemical composition determined with treatments 8 and 10? They are not significantly different in yield.

Author Response

Manuscript ID: foods-1652443

We would like to thank the reviewer of our paper for their helpful feedback to the improvement of this manuscript. We have taken into account all the concerns raised and we have made the suggested modifications.

Reviewer 3 Report

This manuscript is well redacted, argued and discussed. Therefore, it is pleasant to read and understand.

Just a few comments: please revise the manuscript regarding italic forms for the scientific names of plants (eg. line 43).

Section 2: please add refs along this section. If the procedures were optimized during this work, the validation should be included.

Section 3: Please add the outputs concerning the optimization of extraction and statistical analysis/validation; equation which explains the best condition and predicted vs experimental optimum point.

Author Response

(The authors gave the same response as above.)

Round 2

Reviewer 2 Report

The authors have addressed the reviewer's concerns.